# Effect of Beetroot Powder Incorporation on Functional Properties and Shelf Life of Biscuits

**DOI:** 10.3390/foods12020322

**Published:** 2023-01-09

**Authors:** Jasmina Mitrevski, Nebojša Đ. Pantelić, Margarita S. Dodevska, Jovana S. Kojić, Jelena J. Vulić, Snežana Zlatanović, Stanislava Gorjanović, Jovanka Laličić-Petronijević, Sonja Marjanović, Vesna V. Antić

**Affiliations:** 1Faculty of Agriculture, University of Belgrade, Nemanjina 6, 11080 Belgrade, Serbia; 2Health House Olea, Karadjordjeva 8, 26000 Pancevo, Serbia; 3Center for Hygiene and Human Ecology, Institute of Public Health of Serbia “Dr. Milan Jovanonic Batut”, Dr. Subotica 5, 11000 Belgrade, Serbia; 4Institute of Food Technologies, University of Novi Sad, Bulevar cara Lazara 1, 21000 Novi Sad, Serbia; 5Faculty of Technology, University of Novi Sad, Bulevar cara Lazara 1, 21000 Novi Sad, Serbia; 6Institute of General and Physical Chemistry, Studentski trg 12–16, 11000 Belgrade, Serbia; 7Medical Faculty of the Military Medical Academy, University of Defense, Crnotravska 17, 11000 Belgrade, Serbia

**Keywords:** biscuits, beetroot powder, polyphenols, flavonoids, betalains, antioxidant activity, dietary fibers, microbiological properties, functional foods

## Abstract

The demand for ready-to-use functional foods is high, which encourages manufacturers to develop new, nutritionally valuable products. As an excellent source of biologically active compounds, beetroot (*Beta vulgaris* L.) is considered to have highly beneficial effects on health. This research aimed to evaluate the impact of replacing spelt flour (**SF**) with 15%, 20% and 25% beetroot powder (**BP**). The physicochemical and functional properties of biscuits baked at different temperatures (150 and 170 °C) were followed at the beginning, and after 3 and 6 months of storage as standard conditions. Moisture content and water activity (*aw*) gave insight into the biscuits’ shelf life. The value of *aw* from 0.35 to 0.56 indicated appropriate storability. Dietary fiber content in fresh biscuits ranged from 6.1% to 7.6%, protein from 9.2% to 8.9% and sugar from 30.6% to 35.9%. The content of betalain, total polyphenols and flavonoids, and antioxidant activity (DPPH, FRAP) increased with beetroot powder content incorporated. A slight decrease of all the mentioned parameters during the storage indicated satisfied retention of bioactive molecules. The content of prevalent phenolic compounds gallic and protocatechuic acid, identified by HPLC, decreased from 22.2–32.0 and 21.1–24.9 in fresh biscuits to 18.3–23.4 and 17.3–20.3 mg/100 g upon six months of storage, respectively. An increase of the L* and a* and a decrease of the b* coordinate values, compared with the control sample without beetroot values, was noticed as well as the expected level of their change during the storage. The obtained results indicated that biscuits enriched with beetroot powder showed a significantly improved functional, nutritional and antioxidant potential during storage.

## 1. Introduction

Beetroot (*Beta vulgaris* L.) is a biennial, herbaceous plant. Beetroots and leaves are valuable sources of dietary fiber, antioxidants, minerals, and vitamins. The root of beet is commonly used as a food source for humans and should be stored under controlled conditions, at a temperature of 0–5 °C, in a dark place with a relative humidity of 96–100%, in order to maintain its quality. Under these conditions, beetroot can be stored for three to five months [1]. Weight loss during storage is tolerated at a maximum of 5–7%, however, higher losses reduce the beetroot’s quality in terms of nutrients [1]. For long-term preservation of nutritional properties, fresh beetroot can be dehydrated at low temperatures (50 °C for 6 h) and ground to a powder, and then kept in well-closed containers. In this form, beetroot can be added to various food products in order to increase their nutritional and health-promoting properties [2].

The demand for ready-to-use functional foods has increased substantially in recent years, encouraging manufacturers to develop new, nutritionally valuable products with health-promoting properties. In addition to reducing obesity, this can also eliminate or slow the onset of many diseases caused by poor nutrition, such as cardiovascular diseases, diabetes, and certain types of cancer [3]. It is undeniable that functional foods containing high fiber content and bioactive components, which contribute significantly to the proper functioning of the entire body, speeding up metabolism and enhancing digestion [3]. Since biscuits have a pleasant taste and compactness, they are ideal for facilitating the availability of nutrients. In addition, they have a lower moisture content than other bakery products, which increases their shelf life [4]. Their “ready-to-eat” form allows them to be readily available and affordable for all occasions, which, along with the variety of flavors, makes them one of the most popular confectionery products for consumers of all ages. Recently, there has been a significant improvement in reducing the fat and sugar content of confectionery products, especially biscuits, and research is underway to further increase the nutritional value of these types of products by using functional raw materials or supplements [5,6,7].

Studies have been published demonstrating the impact of using different supplements, such as blueberry and blackberry [7], apple [6,8], and carrot [9], to biscuits, and explaining how this influenced their nutritional characteristics. There were also various forms of beetroot used as an ingredient in bakery products, such as beetroot juice which was replaced with part of the water in bread dough, which caused significant positive changes in crust color, crumb porosity, flavor, and color of the bread [10]. In addition to its raw form, beetroot juice, powder, extracts, and residues from processing (pomace) can be used as sources of bioactive compounds. The presence of betalains, phenolic compounds and functional additives derived from beetroot can also contribute to increasing the shelf life of food products, as well as potentially substituting synthetic additives with natural ones. A study conducted by Fernandez et al. [11] described biscuits containing different proportions of dried beetroot leaf powder (up to 12%) [11]. The study’s showed that the biscuits contained more proteins, fibers, minerals, plant pigments, omega-3 fatty acids, and numerous vitamins than no-leaf counterparts [11]. According to Krejcov et al. [12], dried beetroot is a healthy alternative to traditional snacks and can be consumed directly as chips [12]. Among the many nutrients found in beetroot [13,14], it provides calcium, potassium, zinc, sodium, vitamin *A*, *B*_6_, *C*, niacin, and folic acid [14], and therefore it is considered to have highly beneficial health effects, including anti-inflammatory and anti-cancer effects [13,14,15]. Additionally, studies have also shown that regular consumption of beetroots in the diet may reduce the symptoms of hypertension, type 2 diabetes, dementia, arteriosclerosis, and kidney disease [16]. According to a study examining the impact of replacing flour with beetroot powder, the number of bacteria, fungi, and molds decreased after three weeks of storage; in terms of their sensory characteristics, the biscuits with 10% of their usual flour content replaced with beetroot were the most acceptable [17].

In the present study, physicochemical, microbiological, nutritional and functional properties of biscuits fortified with different proportions of beetroot powder were investigated. Beetroot powder was added in different proportions to biscuits in order to enhance antioxidant properties and increase dietary fiber content. Two series of biscuit samples were prepared at different temperatures (150 and 170 °C), with beetroot powder replacing up to 50% of the spelt flour in the recipe. Thus, the percentage of beetroot powder in the total baking mixture was 15%, 20% and 25%. A control sample without beetroot powder was also prepared in every batch using only spelt flour. Both batches of samples were tested for physicochemical properties, focusing on parameters such as moisture content and water activity (*aw*), in order to provide insight into the shelf life of beetroot-containing biscuits. In addition to dietary fiber and protein content, the sugar content was also examined. After baking, as well as after three and six months of storage, microbiological properties of biscuit samples were determined, and the biscuit color change during storage was evaluated. At the same time intervals, betalains, total polyphenols and flavonoid contents, together with antioxidant activity were investigated. Furthermore, individual phenolic compounds were identified and quantified using HPLC.

## 2. Materials and Methods

### 2.1. Chemicals

In this study, *n*-hexane, methanol, Folin–Ciocalteu’s reagent, sodium carbonate, sodium nitrite, aluminum chloride, sodium hydroxide, 2,2-diphenyl-1-picrylhydrazyl (DPPH), acetonitrile, formic acid, epicatechin, catechin, gallic, protocatechuic, *p*-coumaric caffeic, chlorogenic, and vanillic acid, 2,4,6-tripyridyl-*s*-triazine (TPTZ), 6-hydroxy-2,5,7,8-tetramethylchroman-2-carboxylic acid (Trolox) were purchased from Sigma-Aldrich (St. Louis, MO, USA). The chemicals and reagents used were all of analytical grades.

### 2.2. Materials

Spelt integral flour (producer: Jevtić, Serbia), virgin coconut oil (producer: Sri Lanka, importer for Serbia Beyond, Niš), cane sugar (country of origin: Paraguay, importer for Serbia: Superfood, Belgrade), baking powder (Dr. Oetker, Bielefeld, Germany), beetroot (bought at the local market in Pancevo, produced in the village of Glogonj, next to Pancevo, Serbia).

#### Preparation of Beetroot Powder (BP)

Beetroot of the Detroit variety was thoroughly washed and the skins were peeled, then they were cut into 1 mm slices. The rings were arranged, without overlap, on dehydrator trays and dried at 52 °C for 24 h to achieve a constant mass of dehydrated beetroot. After cooling for 3 h, the dried rings were ground in a blender that is specifically designed for grinding grains (VITA-MIX CORP, 1200W, Cleveland, OH, USA). The powder was transferred to PVC containers, closed, and kept away from light and moisture.

### 2.3. Physical Properties of Beetroot Powder and Spelt Flour

#### 2.3.1. Bulk Density

A bulk density was determined according to the procedure [18]. Briefly, in 50 mL beakers, 10 g of sample were placed, and tapping was used to compact the material until maximum volume reduction was achieved. The volume of the beaker was read directly, and the results were expressed as g/mL.

#### 2.3.2. Water-Binding Capacity

The water-binding capacity was determined using a custom method [19]. In a 50 mL centrifuge cuvette, 1 g of the sample was mixed with 30 mL of distilled water. All suspensions were centrifuged at 4000 rpm for 20 min after being left at room temperature for 24 h. After removing the supernatant, the residue was weighed. Water-binding capacity is defined as the mass of bound water per gram of sample.

#### 2.3.3. Oil-Binding Capacity

In a 50 mL centrifuge cuvette, 1 g of the sample was mixed with 10 mL of oil to determine the oil-binding capacity. The mixtures were left at room temperature for 24 h and then centrifuged for 20 min at 4000 rpm. A supernatant was removed, and the residue was weighed. Oil-binding capacity is expressed as a gram of oil per gram of sample. Sunflower oil (Diamond, Zrenjanin, Serbia) was used in the oil-binding process, as well as unrefined coconut oil that had also been used in the manufacture of biscuits.

#### 2.3.4. Swelling Capacity

To determine swelling capacity, 1 g of the sample was mixed with 30 mL of distilled water in a 50 mL centrifuge cuvette. The volume occupied by the dry sample was measured after 24 h of equilibration and is expressed as volume per gram of the dry sample [20]. Swelling capacity (mL/g) = Volume occupied by sample (mL)/mass of original sample (g).

#### 2.3.5. Hydration Capacity

In a graduated beaker filled with distilled water, 0.5 g of the sample was added slowly so that adhering particle would not adhere to its walls. The mixture was allowed to stand for 15 min. A difference was measured between the volume of water before and after the addition of the sample, expressed as g/mL [20]. Hydrated density = sample weight (g)/water level displacement (mL)

### 2.4. Preparation of Biscuits (BPB)

The biscuits are prepared by replacing a certain percentage of whole meal spelt flour (**SF**) with beetroot powder (**BP**). The biscuits were prepared in two batches and baked at 150 °C and 170 °C. For each batch, a blank test was prepared without beetroot powder. The ingredients are listed in Table 1. The images of analyzed biscuits with different **BP** substitution levels are presented in Figure 1.

Beetroot powder and integral spelt flour were mixed in certain proportions, and then 60 g of softened extra virgin coconut oil, 50 g of cane sugar, 40 g of water, and baking powder were added. All the ingredients were mixed well by hand until a consistent mass was achieved. Using a round mold with a diameter of 4.7 cm, the cookies were cut after the mixture rested for half an hour.

The sample was baked for 12 min at 150 °C and 170 °C. After baking, they were allowed to cool in the tray for two hours and then arranged in PET containers with lids and stored without any exposure to light or moisture. Samples were labeled as follows: **A1** (blank sample, 0% beet powder, biscuit baked at 150 °C), **A2** (blank sample, 0% beet powder, biscuit baked at 170 °C), **B1** (15% beet powder, biscuit baked at 150 °C), **B2** (15% beet powder, biscuit baked at 170 °C), **C1** (20% beet powder, biscuit baked at 150 °C), **C2** (20% beet powder, biscuit baked at 170 °C), **D1** (25% beet powder, biscuit baked at 150 °C), **D2** (25% beet powder, biscuit baked at 170 °C). All biscuits were baked in a multi-level oven (every type of the biscuit was on an independent, separate tray, in sun; 20 biscuits on one tray). All four types (**A1**, **B1**, **C1**, **D1**) were baked simultaneously at 150 °C. The same procedure was performed for samples **A2**, **B2**, **C2** and **D2**, each being baked at 170 °C. Given the size of the biscuits (5 cm in diameter), 20 biscuits fit in the tray. Sampling was performed by taking biscuits from the middle of the tray.

### 2.5. Determining the Color of Biscuits

Color measurements were performed ten times using a Minolta Chroma Meter CR-400 colorimeter (Konica Minolta Sensing Inc., Osaka, Japan) adapted for measuring this type of sample. The colorimeter was calibrated before measurement using a white color standard. The results are presented according to the CIELAB color system, where the coordinates are defined as follows: L* is the lightness coordinate of the color (where 0 means black and 100 white), a* is the proportion of red/green color (where a* > 0 means red and a* < 0, indicates green color) and b* is the proportion of yellow/blue color (where b* > 0 indicates yellow and b* < 0 indicates blue color).

### 2.6. Determination of Water Activity (aw) of Starting Raw Materials and Biscuits

Gravimetric moisture analysis was performed in a temperature-controlled dryer at 105 °C to determine constant weight. A LabSwift-aw, Novasina AG, Switzerland A_W_-meter was used to measure the water activity of the samples at 25 °C.

### 2.7. Proximate Composition of Beetroot Powder, Spelt Flour, and Biscuits

The fat protein and ash content were determined using appropriate AOAC methods [21]. Sucrose, D-glucose, and D-fructose were determined by a spectrophotometric method using the enzymatic assay kit R-biopharm (R-BIOPHARM AG, Darmstadt, Germany). The assay was performed according to the instruction manual of the kit producer. The D-glucose concentration was determined before and after the hydrolysis of sucrose by β-fructosidase (invertase). The amount of NADPH formed in this reaction was stoichiometric with the amount of D-glucose. The NADPH was measured by the increase in absorbance at 340 nm. The absorbance was read at 340 nm on the Evolution 201 spectrophotometer, Thermo Scientific. The D-fructose content of the sample was determined subsequent to the determination of D-glucose, after isomerisation by phosphoglucose isomerase. The amount of NADH formed in this reaction was proportional to the amount of D-fructose concentration and was measured by the increase in absorbance at 340 nm. The sucrose content was calculated from the difference in D-glucose concentrations before and after hydrolysis by β-fructosidase.

AOAC method 991.43 [22] was used to determine the total, soluble and insoluble dietary fiber content [2]. Dried food samples were enzymatically degraded with heat-stable α-amylase, protease, and amyloglucosidase. After filtration, the residue was used to determine insoluble dietary fiber, and filtrate, in which were added four volumes of 95% ethanol was used to determine soluble dietary fiber. SDF and IDF residues were corrected for protein, ash and blank for the final calculation of SDF and IDF values.

### 2.8. Spectrophotometric Determination of Betalain, Polyphenol, and Flavonoid Content

#### 2.8.1. Extract Preparation

In order to remove fats, each of the analyzed samples (5 g) was extracted with *n*-hexane (3 × 20 mL) at room temperature for 45 min. Then, the defatted samples were air-dried for 24 h to evaporate organic solvent residues [23]. Afterward, the samples (1 g) were treated three times with 15 mL of methanol for 45 min, centrifuged for 10 min, and the collected supernatants were combined in the flask and filled up with methanol to obtain the final volume of 10 mL.

#### 2.8.2. Determination of Total Betalains

To determine the total betalain content, spectrophotometry was used to determine betanin and vulgaxanthin-I concentrations as two of the main betacyanins and betataxanthins [24]. For betanin and vulgaxanthin-I, the absorption coefficient A^1%^ represents the absorption value of a 1% solution (1 g/100 mL) and for betanin, the value is 1120; for vulgaxanthin-I, the value is 750. The absorption maximum of betanin is at 538 nm, while the emission maximum of vulgaxantin-I is at 476 nm. Betanin also absorbs light at 476 nm, which contributes to the measured value of absorption at this wavelength, so its value needs to be corrected to reflect the amount absorbed by betanin. Betanin absorbs light at 476 nm according to its concentration, and therefore a ratio A_476_/A_538_ is used to calculate its absorption. The calculations are based on the assumption that this ratio is 3.1 at a pH of 6.5. The betanin concentration was determined by measuring the absorption at 538 nm and correcting this value for the absorption at 600 nm (colored impurities absorb at this wavelength). When a solution’s absorbance at 538 nm is between 0.4 and 0.5 AU, it is assumed that the A_538_/A_600_ ratio is 11.5. As a result, solutions are prepared so that their absorption lies within this range. The betalain (betacyanins and betaxanthins) pigment content of the extract was measured using a UV-1800 spectrophotometer (Shimadzu, Kyoto, Japan). The 535 and 476 nm wavelengths were used for betacyanin and betaxanthin analysis, respectively. The betacyanins content was expressed as mg betanin equivalents per g, and the betaxanthins content was expressed as mg vulgaxanthin I equivalents per g [24].

#### 2.8.3. Determination of Total Polyphenols Content

The total polyphenols content (TPC) of analyzed samples was determined using the standard spectrophotometric Folin–Ciocalteu method with slight modifications [23]. Briefly, 0.5 mL of defatted sample extract was mixed with 30 mL of distilled water and 2.5 mL of Folin–Ciocalteu reagent. After 5 min, 7.5 mL of 7.5% Na_2_CO_3_ was added, and the mixture was filled with distilled water up to 50 mL. After incubation of 1 h in the dark, the absorbance was measured at 760 nm against the blank solution. The results are expressed as mg gallic acid equivalents per gram of the sample (mg GAE/g).

#### 2.8.4. Determination of Total Flavonoid Content

The total flavonoid content (TFC) of analyzed extracts was determined spectrophotometrically according to the standard aluminum chloride method with slight modifications [23]. In summary, 0.5 mL of the analyzed sample was mixed with 2 mL of distilled water and 150 µL of 5% NaNO_2_. After 5 min, the solution was treated with 150 µL of 10% AlCl_3_ and shaken up. After 5 min, 1 mL of 1 M NaOH was added and the solution was filled up with distilled water up to 5 mL, shaken up and the absorbance was measured at 510 nm against the blank sample. The results were expressed as mg catechin equivalents per gram of the sample (mg CE/g).

### 2.9. Estimation of Antioxidant Determination

#### 2.9.1. DPPH Radical Scavenging Assay

The antioxidant activity of analyzed samples was estimated using a 2.2-diphenyl-1-picrylhydrazyl (DPPH) radical scavenging assay described by Brand-Williams et al. [25]. Shortly, 200 µL of analyzed samples were mixed with 2 mL of 150 µM DPPH solution, and the mixture was shaken vigorously. The tubes were kept in the dark for 45 min, and afterward, the absorbance was measured at 515 nm. The DPPH scavenging activity was expressed as a percentage of inhibition and calculated using the following formula: Inhibition (%) = [(A_control_ − A_sample_)/A_control_] × 100

#### 2.9.2. Ferric Reducing Antioxidant Power (FRAP)

The ferric-reducing antioxidant power (FRAP) assay was performed by the method described previously [26]. For this purpose, the FRAP reagent was prepared by mixing 2.5 mL of 10 mM 2,4,6-Tripyridyl-s-triazine (TPTZ), 2.5 mL of 20 mM FeCl_3_ × 6H_2_O and 25 mL of 300 mM acetate buffer (pH 3.6). In summary, 100 µL of the analyzed sample was mixed with 300 µL of distilled water and 3 mL of freshly prepared FRAP reagent. The solutions were shaken up and kept in the dark for 40 min, and then the absorbance was read at 593 nm. The results are expressed as mM Trolox equivalents per gram of the sample (mM TE/g).

### 2.10. Qualitative and Quantitative Determination of Polyphenols by HPLC-UV-Vis Method

The polyphenolic compounds were quantified using high-performance liquid chromatography (HPLC) using a Shimadzu Prominence device (Shimadzu, Kyoto, Japan) equipped with LC-20AT binary pumps, CTO-20A thermostats, SIL-20A automatic dispensers, and SPD-20AV UV/Vis detectors. Different wavelengths of light were used to record chromatograms: 280 nm, 320 nm, and 360 nm. Separation was conducted using a Luna C-18 RP column, 5 μm, 250 × 4.6 mm (Phenomenex, Torrance, CA, USA), which was protected by a C18 pre-column, 4 × 30 mm (Phenomenex, Torrance, CA, USA). As the mobile phase, the following solvents were used: A (acetonitrile) and B (1% formic acid) at the flow rate of 1 mL/min and using the following linear gradient: 0–10 min, from 10 to 25% A, then 10–20 min to 60% A, then 20–30 min to 70% A. After the column has been equilibrated to initial conditions, 10% A in 10 min, there is an additional 5 min stabilization period. Filters with a pore size of 0.45 μm were used to filter all samples and solvents before analysis (Millipore, Bedford, MA, USA). The LC Solution Software (Shimadzu, Kyoto, Japan) was used to identify and quantify the peaks obtained.

### 2.11. Determination of Microbiological Correctness

The microbiological correctness of all biscuit samples and a sample of wholemeal flour and dried beetroot powder were evaluated. Test parameters were determined according to the rules for general and special rules of food hygiene at any stage of production, processing, and circulation (Official Gazette of the Republic of Serbia No. 72/10, 62/18) [27]. For testing the parameter Yeasts and molds, the following method was used: SRPS ISO 21527-2:2011, for *Entero bacteriaceae*, SRPS EN ISO 21528-2: 2017, Total number of aerobic mesophilic bacteria (cfu/g): SRPS EN ISO 4833-1: 2014, *Salmonella*, SRPS EN ISO 6579-1: 2017, without Annex D, *Escherichia coli*, SRPS EN ISO 16649-2: 2007, and *Bacillus cereus*: SRPS EN ISO 7932: 2009 [28].

### 2.12. Statistical Analysis

Results obtained in this experiment were analyzed using the IBM SPSS Statistics 25 program (IBM, Armonk, NY, USA). The sample size was shown and descriptive statistics (mean, median, minimum, and maximum) were calculated. Variability measures and standard deviations were calculated for the respective sample groups. The Kruskal–Wallis non-parametric analysis of variance was used to assess the significance of differences between traits measured in cookies during six months (right after production, after three and six months) as well as between cookies with four different contents of beetroot powder (0%, 15%, 20%, 25%). Following a significant result from a Kruskal–Wallis analysis, Dunn’s procedure was implemented as a post hoc multiple comparison procedure to find non-parametric pair wise differences between biscuit traits. Wilcoxon–Mann–Whitney test (a non-parametric analog to the independent samples *t*-test) was used to assess significance of differences between physicochemical properties of beetroot powder and spelt flour. *P*-values of the test are shown for each measured trait. The level of significance was set at 0.05.

## 3. Results and Discussions

Beetroot powder obtained by dehydration of fresh beetroot and subsequent grinding, with no preservatives or additives, was used in this study to produce enriched biscuits with potentially improved functional properties. Significant physical properties and chemical composition of raw materials and prepared biscuits were determined. Further, changes in total betalain, phenolic, and flavonoid content, antioxidant ability, qualitative and quantitative phenolic composition, and changes in visual color parameters of BPBs stored for six months were examined. Biscuit shelf life was estimated based on microbiological safety and water activity values. The biscuits were stored in plastic bags without exposure to light.

### 3.1. Physicochemical Properties of Beetroot Powder, Spelt Flour, and Beetroot–Enriched Biscuits

The value of the bulk density, water-binding capacity, oil-binding capacity, swelling capacity, and hydration capacity are shown in Table 2 for beetroot powder and spelt flour. Research conducted by Kohajdova et al. [29] examined the effect of beetroot powder on the water absorption power of roll dough, and it was determined that the dough was more stable and water absorption improved [29]. In our study, the high water-binding capacities (2.00 g for **BP,** and 7.12 g for **SF**), as well as oil-binding capacities (3.56 g for **BP,** and 3.42 g for **SF**) were found (Table 2). These findings can be attributed to the significant amount of dietary fiber present in both spelt flour and beetroot powder. Therefore, **BP** can be used instead of synthetic additives in food products to increase viscosity, thicken the product and prolong its freshness. Other research has shown similar results when using apple pulp as a functional additive [8]. Significant differences between beetroot powder and spelt flour physicochemical properties were found for bulk density and water-binding capacity (significantly higher values were observed for spelt flour) (Table 2).

Swelling capacity can be used to identify foods rich in dietary fiber [30]. For **BP**, the swelling capacity is 32.5 mL/g, while for **SF**, it is 33 mL/g. These values are significantly highe than those of commercial preparations rich in fiber, such as oats and rice bran, dates, and apple and pear pomase [31]. As a result of the chemical composition of each plant material, differences in swelling capacity were observed [32].

Table 3 represents the chemical compositions of beetroot powder, spelt flour, and biscuit samples. biscuit samples prepared at 150 °C were used for determination of the chemical composition. The temperature was chosen based on preliminary tests of the sensory properties of biscuits, which showed that the samples baked at 150 °C were more sensory acceptable, which will be described in more detail in our subsequent publication. Compared to beetroot powder, spelt flour contained a higher level of fat (2.41% vs. 0.59%). Furthermore, beetroot powder contained a higher level of dietary fiber and ash than spelt flour due to the higher mineral content of beetroot [33]. In terms of protein, spelt flour had 20% more protein than beetroot powder. The ash content of beetroot powder was 3.8%, indicating that beetroots were a good source of minerals. On the other hand, beetroot powder had a moisture content of 6.8%, indicating a good storage capacity. Lucky et al. [34] reported similar results with beetroot powder samples, with 3.57% ash [34].

Table 3 shows that the proportion of carbohydrates was very similar for all biscuit samples, ranging from 58.39% (sample **B1**) to 59.95% (sample **A1**). As was expected, the protein content of the samples slightly decreased with increasing beet content from 9.17% (sample **B1**) to 8.94% (sample **D1**). A similar result was obtained by other authors, and it was observed that the protein content was 12.42%, with an increase in the share of beetroot up to 25% [13]. All biscuit samples had a fat content of approximately 25% (Table 3). Furthermore, the ash content increased from 2.75% (sample **B1**) to 2.95% (sample **D1**), and the moisture content decreased from 4.15% (sample **B1**) to 3.84% (sample **D1**), which may be explained by the relatively high value of the water-binding capacity.

In addition, the highest level of dietary fiber in sample **D1** (7.60%) was observed due to its high proportion of **BP**, and it decreased proportionally with the decline in beet content (Table 3). A similar result was obtained by Nazni et al. [35], in which the amount of ash and fiber in the samples increased withthe increase in beetroot powder share [35]. Dietary fiber is recognized as a highly desirable ingredient in food products because, among other effects, it reduces the risk of diabetes [36].

### 3.2. Changes in the Total Content of Betalains, Polyphenols, and Flavonoids

In this study, changes in the content of betalains, polyphenols, flavonoids, and antioxidant properties were monitored over 6 months of biscuits storage under normal conditions, at room temperature.

#### 3.2.1. Total Betalains

Betacyanins and betaxanthins concentrations in beetroot powder as well as in biscuits baked at 150 °C and 170 °C, are listed in Table 4. At constant storage conditions, betaxanthins content decreased by 30%, whereas betacyanins content decreased by 36%. This indicates that betaxanthins are slightly more stable during storage than betacyanins. As expected, the amount of betacyanins and betaxanthins in the biscuit increased with an increase in the proportion of dried beetroot (Table 4). The stability of betalains during storage is affected by temperature. It has been reported that the degree of degradation of betalains increases with increasing temperature [37], as confirmed in our study (Table 4). The most significant loss of betacyanins during storage at room temperature was observed in the samples containing 25% beetroot powder (**D**). In biscuit sample **D1**, baked at 150 °C, 70% of betacyanins degraded after three months, and in sample **D2**, baked at 170 °C, 52% of betacyanins were lost after three months. Results showed that samples baked at a lower temperature had a higher initial betacyanins value, but a greater statistically significant loss (*p* ˂ 0.05) during storage (57–70%), whereas samples baked at a higher temperature had a lower initial betacyanins value and a smaller loss over time (16–52%). Furthermore, it was also observed that samples of biscuits baked at lower temperatures had a higher initial value of betaxanthins, but also significantly higher loss of betaxanthin (*p* ˂ 0.05) during storage (54–71%), whereas samples baked at higher temperatures had a lower initial value of betaxanthins, but their loss over time was also lower (27–53%). According to the results of these studies, betalains in biscuits appear to be unstable during long-term storage at room temperature. However, their loss is lower in products baked at higher temperatures, which may be related to moisture content [38] and is correlated with lower water activity values (Section 3.6). Water activity significantly affects betalain stability. After beetroots are subjected to methods to reduce water activity, such as concentration and drying, betacyanins stability increases. According to Henriette [37], the most pronounced degradation occurs at *aw* = 0.64 for encapsulated beet pigments [37].

#### 3.2.2. Total Polyphenols and Flavonoid Content

Many studies have shown that red beetroot is a rich source of phenolic compounds, including phenolic acids and flavonoids, and is therefore considered to have various health benefits [39]. Namely, these biologically active substances can scavenge free radicals, which can cause many pathological processes in cells leading to cancer cell formation, inflammation and cardiovascular diseases [40,41]. The evaluation of total polyphenols and total flavonoid content in the biscuits samples during storage was performed spectrophotometrically, and the results are shown in Figure 2a,b (TPC at 150 °C and 170 °C, respectively) and Figure 3a,b (TFC at 150 °C and 170 °C, respectively). The polyphenols and flavonoid content largely depend on the type of extraction solvent used. According to the literature data, alcoholic extraction agents, such as methanol and ethanol, are the most efficient for extracting phenolic compounds from these types of samples [42].

The results showed that total polyphenols content ranged from 9.08 to 34.14 mg GAE/g in the biscuits samples baked at 150 °C, between 9.24 and 33.94 mg GAE/g for those baked at 170 °C. It can thus be concluded that total polyphenol content depends on the beetroot contents in the biscuits. Furthermore, increasing the beetroot share resulted in higher total polyphenols content in all samples. Moreover, replacing 15% of spelt flour with the same quantity of beetroot powder resulted in a significant increase in polyphenol content in biscuit samples baked at both applied temperatures. The trend continued with a further increase in the portion of beetroot (up to 20%) compared with the sample containing 15% beetroot powder. However, there was no statistically significant difference in polyphenol content in biscuit samples with a 25% beetroot powder content (**D1**: 37.71, 35.98, 34.14 mg GAE/g; **D2**: 38.02, 36.29, 33.94 mg GAE/g, start of storage, after 3 and 6 months, respectively), compared to those with 20% (**C1**: 35.88, 33.94, 32.41 mg GAE/g; **C2**: 37.71, 36.18, 34.35 mg GAE/g, start of storage, after 3 and 6 months, respectively) of this functional ingredient. Additionally, it was observed that total polyphenol content decreased during storage at room temperatures. The most significant decrease of 14% in polyphenol content during six months of storage was observed in the biscuit sample enriched with 15% beetroot baked at a higher temperature (**B2**), while the lowest reduction of 9% was recorded in the sample containing 20% beetroot baked at 170 °C (**C2**). Interestingly, in most cases, the biscuit samples baked at 170 °C had a higher total polyphenol level than those baked at 150 °C with the same portion of beetroot powder. However, the differences obtained in samples with the same percentage of beetroot, baked at different temperatures, were not statistically significant (*p* < 0.05). After statistical data processing, it was determined that the content of polyphenols in samples **A1, A2** and **D1, D2** during storage was statistically significantly different (*p* ˂ 0.05). Additionally, the analysis found that there were no significant differences in all three storage periods only between samples **C1, C2,** and **D1, D2**, while significant differences (*p* ˂ 0.05) in polyphenol contents were found between samples **B1, B2** and **C1, C2** as well as **B1, B2** and **D1, D2**. The differences obtained in samples with the same percentage of beetroot, baked at different temperatures, were not statistically significant.

The contribution of powdered functional ingredients to biscuits’ polyphenol content has recently been reported in the literature [43,44,45,46]. Cookies prepared by replacing 15% wheat flour with banana peel and bee pollen powder exhibited a TPC of 0.921 and 4.01 mg GAE/g, respectively [43,44]. Furthermore, the TPC of 3.07 mg GAE/g was observed in cookie samples with 10% apricot as a functional ingredient [45]. The results for total polyphenols content in our study cannot be directly compared to literature data due to the differences in ingredients used (fruits, vegetables, flours), their amounts while preparing the samples, as well as the baking temperature. However, it may be concluded that beetroot powder, as a functional ingredient in biscuits, has a strong impact on total polyphenol content, significantly stronger than the other functional ingredients described in the literature.

Flavonoids are an important class of natural products and represent a significant index for nutritional evaluation of food ingredients due to their antioxidant activity [47]. The total flavonoid content in biscuit samples was determined using the aluminum chloride method and the results ranged from 2.93 to 16.57 mg CE/g. The highest value was observed in the sample with 25% beetroot powder baked at 170 °C (Figure 3b). Again, similar to TPC, the results indicated that flavonoid content depends on the portion of beetroot powder in the analyzed samples and increases with its increase. Significant differences were observed between three time period of storage in all type of biscuits. During six months of storage amount of total flavonoids decreased significantly (Figure 3a,b). The differences obtained in samples with the same percentage of beetroot, baked at different temperatures, were not statistically significant.

### 3.3. Antioxidative Determination during Storage

#### 3.3.1. DPPH Assay

In this study, the potential of the tested samples to scavenge free radicals was determined by measuring their ability to reduce 2,2-diphenyl-1-picrylhydrazyl (DPPH) radicals [25]. The results presented in Figure 4a,b show that the highest antioxidant activity was observed in pure beetroot powder, which showed 90% inhibition of free radicals, with slight variations during six months of storage. It was found that the **D2** sample had a high antioxidant activity of 68.60%, with 25% beetroot powder, which is associated with beetroot itself, which has an exceptional antioxidant activity. Same as for the pure beetroot powder, the results varied very little during storage. The samples **A1** and **A2**, which do not contain beetroot powder, exhibited the lowest antioxidant activity, 15.76%. A trend of decreasing antioxidant activity was observed in both series of samples, prepared at different temperatures, as well as with the decrease in the proportion of beetroot in the biscuit sample in each series. The lowest antioxidant activity was observed in biscuits without beetroot powder (**A** type) and the highest activity in biscuits with 25% of beetroot powder (**D** type) (Figure 4a,b). The baking temperature alone did not significantly affect the DPPH radical activity, but a different ratio of the ingredients in the biscuit may result in different antioxidant activity. In addition to their concentration, antioxidants have a synergistic action, which is determined by their structure as well as by their interaction. Literature results have also shown the effect of functional cookie ingredients on their antioxidant activity. The cookies enriched with 15% moringa leaf powder showed 15.25% DPPH inhibition compared with 1.56% of the control sample [48], while maqui berry powder in portions from 2.5% to 10% significantly increased the antioxidant potential of cookies. Furthermore, the study conducted by Juon et al. [49] demonstrated that cookies prepared by substituting 15% of wheat flour with banana peel powder exhibited DPPH inhibition of 70.30% [49]. Additionally, cookies with cranberry [50], acai berry [51] and blueberry powder [52] showed a similar increase in the antioxidant capacity [50,51,52].

#### 3.3.2. FRAP Assay

The FRAP method consists of reducing yellow Ferro-tripyridyltriazine complexes (Fe(III)-TPTZ), at a pH value of 3.6, in the presence of electron-donating antioxidants, into intensely blue ferro complexes (Fe(II-TPTZ). A FRAP unit is defined as the amount of antioxidants required to reduce 1 mole of Fe(III) to Fe(II). According to the DPPH method, antioxidants are able to bind free radicals, while according to the FRAP method, antioxidants are able to reduce iron ions [26]. There is an electron-exchange mechanism involved in both reactions.

The results of antioxidant activity assessment in analyzed samples using FRAP assay are presented in Figure 5a,b. Obtained results indicated that the amount of beetroot powder directly affects the antioxidant capacity of the biscuit samples. The samples with no added beetroot powder had the lowest reducing ability of 0.02 mM TE/g baked at both temperatures used. Substituting 15% of spelt flour with the same proportion of beetroot powder increased activity by approximately 16 times. By further increasing the proportion of beetroot to 20% and 25%, the reduction capacity of 0.45 and 0.50 mM TE/g was achieved, respectively. Additionally, it was observed that antioxidant potential decreased during storage at room temperature. On the other hand, a high positive correlation was observed between the results of DPPH and FRAP methods, *R*^2^ = 0.9351 and 0.8654 at 150 °C and 170 °C baking temperatures, respectively. This indicates that these analytical methods do not show significant deviations in evaluating the antioxidant activity of analyzed samples, which is in agreement with the findings obtained by Thaipon et al. [53].

### 3.4. Changes in Qualitative and Quantitative Phenolic Composition during Storage

Results of qualitative and quantitative HPLC analyses of beetroot powder, spelt flour and biscuits are presented in Table 5 and Table 6. HPLC analysis of beetroot powder showed the presence of epicatechin, catechin, gallic, protocatechuic, *p*-coumaric, caffeic, chlorogenic and vanillic acid. Gallic, protocatechuic, caffeic, chlorogenic and vanillic acids were detected in spelt flour. The levels of epicatechin, catechin, and *p*-coumaric acid were below the limit of detection. As presented in Table 5, only gallic, protocatechuic and vanillic acids were continuously decreasing during storage, while epicatechin, catechin and chlorogenic acid showed an increase after three months, and then a decrease in the sixth month, but above the initial level. There was a sudden drop in the concentration of *p*-coumaric and caffeic acid in the third month of storage, followed by an increase in concentration in the sixth month. A quantitative analysis indicated that the concentration of phenolic compounds in dried beetroot decreased by approximately 35% during storage. Spelt flour, containing approximately ten times less polyphenols than beetroot powder, showed a gradual decrease in the concentration of each compound over time. Polyphenols of spelt flour loss was 14% after six months of storage. At the beginning and after three months of storage of a blank test (biscuits without beetroot, samples **A1** and **A2**), epicatechin, catechin, and *p*-coumaric acid were not detected, but after six months catechin was detected, for both baking temperatures. In our study, the content of some phenolic compounds, such as epicatechin, catechin, as well as *p*-coumaric acid increased, probably because the biscuit production steps destabilised their links with some molecules, e.g., arabinose and short arabinoxylan chains (conjugated compounds), or cellulose and lignin (bound compounds) [54]. A similar observation was published in the study by Hidalgo et al. [55], where control water biscuits were prepared from bread wheat flour, and which was substituted partially with sprouted-cereal. Furthermore, Abdel-Aal and Rabalski [56] observed an increase in free phenolics while baking whole meal bread, biscuits and muffins. Moreover, Fares et al. [57] noticed a consistent increase in bound phenolic acids during pasta cooking. There was a decrease in polyphenolic compounds in sample **B1** (15% beetroot) over time (74.40 to 59.59 mg/100g), whereas, in sample **B2**, a slight increase was observed from 42.80 to 54.51 mg/100g. In **B2** gallic, protocatechuic and caffeic acid concentrations significantly increased during storage (Table 6). Compared with the initial value, samples with 20% beet, **C1**, and **C2**, as well as 25%, **D1**, and **D2**, showed a statistically significant (*p* ˂ 0.05) decrease in total polyphenols in both the third and sixth months of storage. The sample containing 25% beetroot, baked at 170 °C (**D2**), contained the highest levels of polyphenols, 110.13 mg/100 g (Table 6).

At the beginning of storage, no samples contained catechins *p*-coumaric acids. After three months of storage, catechin was first detected in biscuits containing beetroot, and after six months also in biscuits without beetroot, similar to the study by Dvořáková et al. [54]. A statistically significant increase in catechin content was observed in biscuits with greater content of beetroot at both temperatures (Table 6). *p*-Coumaric acid was detected in biscuits containing beetroot baked at 150 °C after three months of storage, while its concentration was below the detection limit after six months of storage. In samples **B2, C1, C2, D1** and **D2**, epicatechin was detectable at the beginning of storage, but by the end of storage, it was below the method’s limit of quantification. In biscuits baked at both temperatures, there were significantly higher concentrations of epicatechin in biscuits with higher content of beetroot (Table 6). Based on these results, it can be concluded that catechin, epicatechin, and *p*-coumaric acid are the most unstable polyphenols in the samples. The study by Georgiev et al. [58] showed that beetroot contained those phenolic compounds and that they were not detected in biscuit samples after six months of storage. Various compounds in the biscuit sample (sugars and vitamin C) may influence change in phenolic content [59]. The most abundant acids found in the samples were gallic, protocatechuic, and vanillic acids, respectively.

### 3.5. Changes in Visual Color Parameters

Color is an important parameter in determining whether the consumer will accept a product. A study was conducted on the color of biscuits containing beetroot powder and the values of L* (lightness), a* (redness) and b* (yellowness) were compared for different proportions of beetroot powder (Table 7). Darker-colored biscuits were produced by substituting 15%, 20% and 25% spelt flour with beetroot powder. While storing biscuits, the value of the L* coordinate increased. This effect was expected to occur due to the inclusion of dark colored compounds [60]. Compared to the control sample without beetroot, the a* coordinate value increased and the b* coordinate value decreased. The L* values of the control samples were 53.91, 55.91 and 56.74 (start of storage, after 3 and 6 months, respectively) for sample **A1**, and 53.14, 56.82 and 55.61 (start of storage, after 3 and 6 months, respectively) for sample **A2**. With increasing baking temperature, the biscuits’ color darkened, which can also be attributed to a lower proportion of betalains and their degradation at higher temperatures. A statistically significant difference (*p* ˂ 0.05) of the L*coordinates of the biscuit was determined in the third month, while there were no statistically significant differences in the sixth month. In the beetroot-containing biscuits, the value of the coordinate L* decreased, with the increase in the proportion of beetroot from **B1** (26.80, 27.72, 30.55), through **C1** (26.57, 26.23, 27.46), to **D1** (25.88, 25.47, 27.16). A statistically significant decrease in the L* coordinate (*p* ˂ 0.05) with the reduction in spelt flour share occurred due to the loss of the flour’s white color [61]. Biscuits without beetroot powder had redness values of **A1** (9.05, 9.45, 9.22) and **A2** (10.04, 9.13, 9.49) while those with beetroot powder had redness values of **B1** (23.70, 24.00, 26.60), **C1** (24.49, 26.12, 26.08) and **D1** (22.13, 24.36, 24.49) (Table 7). This indicates that in biscuits with beetroot powder, the proportion of red color increased statistically significantly (*p* ˂ 0.05). With regard to the yellow color and the b* coordinate, it statistically significantly (*p* ˂ 0.05) increased in the control samples **A1**, **A2**, and decreased in the beetroot samples in the sixth month. Uthumporn et al. [62] observed that the value of the L* coordinate decreased, as the proportion of dietary fiber in the product increased. Color differences may be caused by uneven exposure of the biscuit surface to the baking temperature or by chemical reactions such as caramelization and Maillard reaction [60]. It is reported that temperature is the most important factor that affects betalains’ stability during storage and processing [63]. However, their stability is also affected by pH, oxygen, light, water activity (*aw*), enzymes and some metal ions [61], which may be the reason for changing the biscuits’ color during storage in our study. Borrelli et al. [64] proved that the reaction between proteins and carbohydrates is responsible for the creation of brown shades, and Nyam et al. [65] demonstrated that the value of the L* coordinate significantly decreases with the addition of rose seed powder [64,65].

### 3.6. Estimation of Biscuits Shelf Life Based on Microbiological Safety and Water Activity

The water activity (*aw*) of confectionery products is used as a critical control point for the development of these products, as it provides information regarding the stability of the sample, the growth of microorganisms on the surface, and the shelf life of the final product. Controlling *aw* prevents the degradation of the product’s structure, texture, and stability. Pitalua et al. [66] concluded that the stability of betalains in microcapsules depended on *aw* [66]. According to the authors, for *aw* values in the range of 0.11–0.52, there were no significant differences in betalain content during 45 days of storage. However, as a result of storage, encapsulates with an *aw* of 0.75 to 0.90 showed significantly reduced betalain concentrations. The value of *aw* ranged from 0.355–0.559 in this study (Table 8), which is more than acceptable for preserving the microbiological, chemical and physical stability of the product. As a result of storage for three months, the value of *aw* further decreased, and all samples reached a value of about 0.35, supporting the possibility of longer storage of biscuits containing beetroot.

The optimal value of *aw* for the growth of a large number of microorganisms is 0.99. Bacteria require the most aw of all microorganisms. Yeast has a lower *aw* requirement, and molds have the lowest requirement. In general, bacteria that cause food spoilage grow when *aw* is greater than 0.91, while molds and yeasts can grow when *aw* is less than 0.80 [67].

Although biscuits are microbiologically stable products with a long shelf life, care should be taken in packaging and storage to prevent mold growth. Microbiological analysis of biscuit samples with added beetroot was performed at the beginning of storage, and then after the third and sixth months of storage under constant conditions. The biscuit samples were analyzed for yeasts and molds, enterobacteria, and aerobic mesophilic bacteria. In addition, spelt flour was tested for yeasts and molds, enterobacteria, *Bacillus* and aerobic mesophilic bacteria, and beetroot powder for *Salmonella*, *Escherichia coli*, *Enterobacteria* and aerobic mesophilic bacteria.

Five trials (*n* = 5) of each biscuit sample were performed. The limit values are expressed in cfu/g and were considered acceptable if they were between *m* (10^2^) and *M* (10^3^ cfu/g). The number of samples that should fall within this interval is indicated in the table with the letter “*c*.”.

During the third and sixth months, the amount of yeast and mold in sample **B2** increased (Appendix A). The results corresponded with those for the spelt flour, where yeasts and molds were present at the same levels (Appendix A). Considering that the water activity of spelt flour and biscuits decreased during storage, and yeasts and mold grew in them (up to 1900 cfu/g), it can be concluded that these organisms originated from spelt flour.

*Enterobacteria* were not detected in the tested biscuit samples (Appendix A).

The content of aerobic mesophilic bacteria is shown in Appendix A. The most prevalent bacterial species in samples **D1** and **D2** was *Bacillus species* in the sixth month (250 out of 350 aerobic bacteria). In samples **A1** and **A2**, aerobic mesophilic bacteria were observed to increase in the third month and then decrease to levels lower than those at the beginning of the study in the sixth month. Research by Marriott et al. [67] suggests that bacterial metabolic by-products and competition for space and food may reduce reproduction to the point where it almost stops or even slightly decreases [67].

Based on the results, it can be concluded that *Salmonella, Escherichia coli,* and *Enterobacter* were not detected in any of the biscuit samples, and the samples were satisfactory according to food safety criteria [28]. Molds and yeasts found in sample **B2** did not affect the overall microbial profile of the final product, so the samples were acceptable according to food safety criteria [28].

## 4. Conclusions

Beet powder’s potential in bridging dietry fiber and antioxidant gaps in modern diets and its applicability in the production of fortified biscuits was demonstrated. The partial replacement of the spelt flour with beetroot powder up to the level of 25% enabled the production of biscuits with a more attractive color, higher polyphenolics, and dietary as well as presence of the betalains. The biscuits with maximal content of beetroot powder incorporated contained the highest concentration of polyphenolics and the fiber content required to claim a high fiber product (7.60%). Good retention of polyphenolics during storage was confirmed. There was no effect of baking temperature on antioxidant activity, while the antioxidant activity decreased slightly over time. However, antioxidant activity remained much higher in control samples even after the storage period. Further research should be conducted to investigate the effect of beet powder on the handling of fresh dough, the sensory properties of biscuits with various amounts of beet powder incorporated as well as the impact of beet powder incorporation on biscuits’ glycemic index and glycemic load. 

## Figures and Tables

**Figure 1 foods-12-00322-f001:**
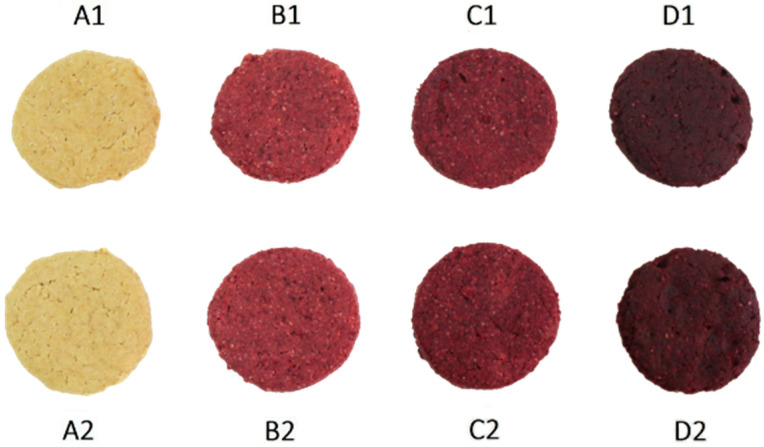
The visual appearance of biscuits with various proportions of beetroot powder prepared according to the formulation given in Table 1.

**Figure 2 foods-12-00322-f002:**
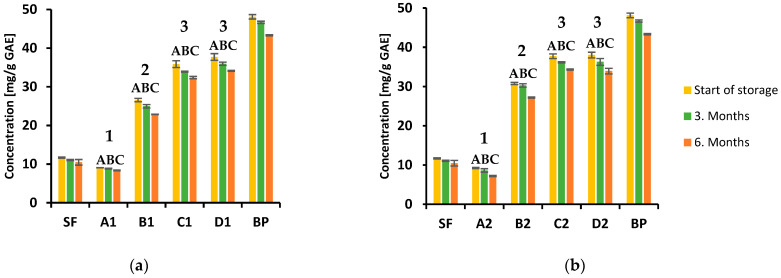
Content of total polyphenols in analyzed biscuits during 6 months of storage at room temperature: (**a**) baked at 150 °C; (**b**) baked at 170 °C. Significant differences in mean values (*p* < 0.05) within one type of biscuits between three time periods are presented with different superscript-letters (A, B, C). Significant differences in mean values between four types of biscuits are presented with different superscript-numbers (1, 2, 3).

**Figure 3 foods-12-00322-f003:**
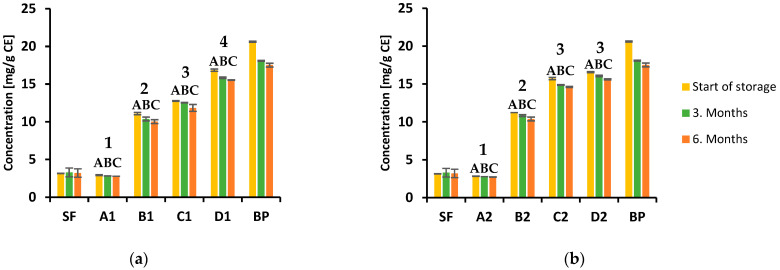
Content of total flavonoid in analyzed biscuits during 6 months of storage at room temperature: (**a**) baked at 150 °C; (**b**) baked at 170 °C. Significant differences in mean values (*p* < 0.05) within one type of biscuits between three time periods are presented with different superscript-letters (A, B, C). Significant differences in mean values between four types of biscuits are presented with different superscript-numbers (1, 2, 3, 4).

**Figure 4 foods-12-00322-f004:**
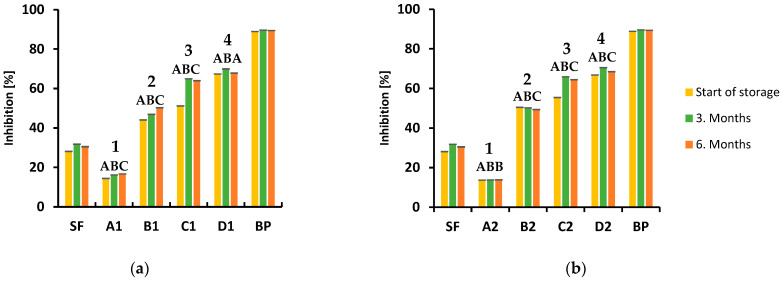
Inhibition of DPPH radicals (%) of analyzed biscuits during 6 months of storage at room temperature: (**a**) baked at 150 °C; (**b**) baked at 170 °C. Significant differences in mean values (*p* < 0.05) within one type of biscuits between three time periods are presented with different superscript-letters (A, B, C). Significant differences in mean values between four types of biscuits are presented with different superscript numbers (1, 2, 3, 4).

**Figure 5 foods-12-00322-f005:**
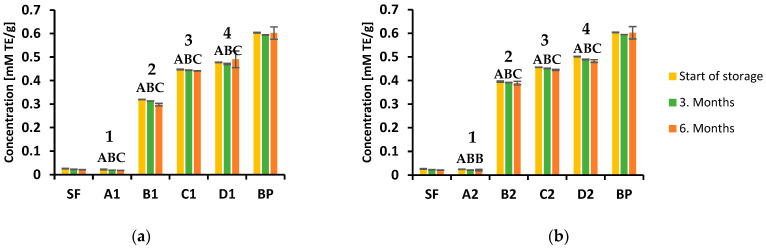
Ferric reducing antioxidant power (FRAP) of analyzed biscuits during 6 months of storage at room temperature: (**a**) baked at 150 °C; (**b**) baked at 170 °C. Significant differences in mean values (*p* < 0.05) within one type of biscuit between three time periods are presented with different superscript-letters (A, B, C). Significant differences in mean values between four types of biscuits are presented with different superscript numbers (1, 2, 3, 4).

**Table 1 foods-12-00322-t001:** Biscuits formulation.

Sample Designation		Spelt Flour (g)	Amount of Beetroot Powder (g)	Mass Fraction of Beetroot Powder (%)
150 °C	170 °C			
**A1**	**A2**	150	0	0
**B1**	**B2**	105	45	15
**C1**	**C2**	90	60	20
**D1**	**D2**	75	75	25

**Table 2 foods-12-00322-t002:** Physicochemical properties (bulk density, oil-binding capacity, water-binding capacity, swelling capacity, hydration capacity) of beetroot powder and spelt flour.

	Bulk Density [g/mL]	Water-binding Capacity [g/g]	Oil-binding Capacity [g/g]	Swelling Capacity [mL/g]	Hydration Capacity, [g/mL]
**Beetroot Powder (BP)**	14.00 ± 0.61 ^A^	2.00 ± 0.12 ^A^	3.56 ± 0.21 ^A^	32.50 ± 1.02 ^A^	11.00 ± 0.44 ^A^
**Spelt flour** **(SF)**	17.00 ± 0.69 ^B^	7.12 ± 0.22 ^B^	3.42 ± 0.19 ^A^	33.00 ± 1.23 ^A^	11.00 ± 0.53 ^A^

Statistically significant differences (*p* < 0.05) between mean values of physicochemical properties between beetroot powder and spelt flours are presented with different superscript-letters (A, B).

**Table 3 foods-12-00322-t003:** Proximate composition of red beetroot powder (**BP**), spelt flour (**SF**), and biscuits samples prepared at 150 °C. Mean and standard deviation are presented.

Proximate Composition (%)	BP	SF	A1	B1	C1	D1
**Fat**	0.59 ± 0.12 ^A^	2.41 ± 0.13 ^B^	25.62 ± 0.35 ^C^	25.55 ± 0.38 ^C^	25.01 ± 0.34 ^C^	25.10 ± 0.30 ^C^
**Total carbohydrate**	77.41 ± 0.60 ^A^	71.89 ± 0.19 ^B^	59.95 ± 0.69 ^C^	58.39 ± 0.62 ^C^	59.11 ± 0.64 ^C^	59.18 ± 0.79 ^C^
**Total dietary fiber**	19.90 ± 0.55 ^A^	4.03 ± 0.21 ^B^	3.36 ± 0.23 ^B^	6.85 ± 0.20 ^C^	7.10 ± 0.21 ^C^	7.60 ± 0.41 ^C^
**Insoluble dietary fiber**	14.80 ± 0.26 ^A^	3.08 ± 0.32 ^B^	2.64 ± 0.14 ^B^	5.06 ± 0.29 ^C^	5.30 ± 0.28 ^C^	5.60 ± 0.39 ^C^
**Soluble dietary fiber**	5.10 ± 0.29 ^A^	0.95 ± 0.14 ^B^	0.71 ± 0.10 ^B^	1.79 ± 0.16 ^C^	1.80 ± 0.19 ^C^	2.04 ± 0.25 ^C^
**Total sugar**	57.06 ± 1.2 ^A^	1.58 ± 0.19 ^B^	22.19 ± 0.32 ^B^	30.63 ± 0.33 ^C^	33.69 ± 0.35 ^D^	35.89 ± 0.53 ^D^
**Sucrose**	51.82 ± 0.5 ^A^	0.51 ± 0.08 ^B^	20.96 ± 0.18 ^C^	28.97 ± 0.22 ^D^	31.97 ± 0.20 ^E^	33.86 ± 0.20 ^F^
**Glucose**	3.46 ± 0.42 ^A^	0.82 ± 0.11 ^B^	0.85 ± 0.09 ^B^	1.14 ± 0.06 ^B^	1.19 ± 0.09 ^B^	1.42 ± 0.21 ^B^
**Fructose**	1.78 ± 0.30 ^A^	0.25 ± 0.04 ^B^	0.38 ± 0.05 ^B^	0.53 ± 0.05 ^B^	0.53 ± 0.06 ^B^	0.61 ± 0.13 ^B^
**Protein**	11.4 ± 0.20 ^A^	13.50 ± 0.11 ^B^	8.92 ± 0.12 ^C^	9.17 ± 0.08 ^C^	9.06 ± 0.13 ^C^	8.94 ± 0.1 ^C^
**Ash**	3.80 ± 0.18 ^A^	1.40 ± 0.07 ^B^	2.54 ± 0.10 ^C^	2.75 ± 0.08 ^CD^	2.92 ± 0.11 ^D^	2.95 ± 0.14 ^D^
**Moisture**	6.80 ± 0.18 ^A^	10.80 ± 0.18 ^B^	2.96 ± 0.14 ^C^	4.15 ± 0.06 ^D^	3.91 ± 0.07 ^D^	3.84 ± 0.18 ^D^

Statistically significant differences (*p* < 0.05) between mean values of **BP**, **SF** and four biscuit types are presented with different superscript-letters (A, B, C, etc.).

**Table 4 foods-12-00322-t004:** Content of betacyanins and betaxanthins in beetroot powder (**BP**) and the samples of biscuits baked at different temperatures (150 °C and 170 °C) during three time periods.

	Total Betacyanins (mg_beanin_/100g)			Total Betaxanthins (mg _vulgaxantin I_/100g)		
	Start of Storage	3 Months	6 Months	Start of Storage	3 Months	6 Months
**B1**	3.19 ± 0.00 ^A;1^	1.47 ± 0.01 ^B;1^	1.27 ± 0.02 ^C;1^	2.23 ± 0.03 ^A;1^	1.41 ± 0.34 ^B;1^	1.01 ± 0.00 ^C;1^
**C1**	3.43 ± 0.01 ^A;1^	2.28 ± 0.00 ^B;2^	1.13 ± 0.01 ^C;2^	3.31 ± 0.71 ^A;2^	2.11 ± 0.02 ^B;2^	0.97 ± 0.01 ^C;1^
**D1**	3.88 ± 0.05 ^A;2^	3.24 ± 0.045 ^B;3^	1.23 ± 0.01 ^C;12^	3.28 ± 0.06 ^A;2^	2.74 ± 0.05 ^B;3^	0.95 ± 0.02 ^C;1^
**B2**	1.78 ± 0.01 ^A;1^	1.27 ± 0.01 ^B;1^	1.15 ± 0.01 ^C;1^	1.65 ± 0.037 ^A;1^	1.18 ± 0.03 ^B;1^	0.92 ± 0.19 ^C;1^
**C2**	1.90 ± 0.05 ^A;1^	1.36 ± 0.04 ^B;12^	1.61 ± 0.02 ^C;2^	1.49 ± 0.036 ^A;2^	1.06 ± 0.02 ^B;2^	1.11 ± 0.02 ^C;2^
**D2**	2.30 ± 0.01 ^A;2^	1.64 ± 0.01 ^B;2^	1.11 ± 0.01 ^C;1^	2.18 ± 0.035 ^A;3^	1.56 ± 0.02 ^B;3^	1.10 ± 0.02 ^C;2^
**BP**	12.79 ± 0.10	10.59 ± 0.16	8.14 ± 0.12	9.17 ± 0.54	8.88 ± 0.26	6.34 ± 0.18

Significant differences in mean values (*p* < 0.05) within one type of biscuits between three time periods are presented in different superscript-letters (*p* < 0.05). Significant differences in mean values between three types of biscuits are presented in different superscript-numbers (A, B, C). Significant differences in mean values between three types of biscuits are presented with different superscript-numbers (1, 2, 3). The baking temperature does not apply to the beetroot powder sample (**BP**).

**Table 5 foods-12-00322-t005:** Qualitative and quantitative composition of polyphenolic compounds present in beetroot powder and spelt flour during storage period.

	Beetroot Powder				Spelt Flour	
Polyhenolic Composition mg/100 g	Start of Storage	3 Months	6 Months	Start of Storage	3 Months	6 Months
Gallic acid	106.36 ± 5.01	79.76 ± 3.87	78.29 ± 1.12	4.28 ± 0.19	4.00 ± 0.19	3.10 ± 0.16
Protocatechuic acid	88.04 ± 3.98	49.17 ± 2.34	32.76 ± 1.01	4.47 ± 0.18	4.33 ± 0.20	3.62 ± 0.20
Epicatechin	17.57 ± 0.69	18.71 ± 0.78	16.92 ± 0.54	n.d.	n.d.	n.d.
Catechin	45.28 ± 2.11	68.05 ± 2.98	48.19 ± 2.01	n.d.	n.d.	n.d.
*p*-Coumaric acid	4.66 ± 0.19	2.63 ± 0.09	4.22 ± 0.08	n.d.	n.d.	n.d.
Caffeic acid	11.47 ± 0.43	4.75 ± 0.21	6.80 ± 0.31	0.58 ± 0.02	0.28 ± 0.01	0.29 ± 0.01
Chlorogenic acid	3.93 ± 0.13	6.28 ± 0.28	5.63 ± 0.21	0.20 ± 0.01	0.19 ± 0.01	0.05 ± 0.01
Vanillic acid	43.62 ± 1.79	30.01 ± 1.43	14.04 ± 0.77	25.92 ± 1.12	23.95 ± 1.13	20.52 ± 1.11
Total	320.93 ^A^	259.37 ^B^	206.85 ^C^	35.44 ^A^	32.74 ^B^	30.45 ^C^

Significant differences in mean values (*p* < 0.05) of total polyphenolic composition between three time periods in beetroot powder as well as in spelt flour are presented as different superscript letters (A, B, C). n.d.—not detected.

**Table 6 foods-12-00322-t006:** Qualitative and quantitative composition of polyphenolic compounds present in biscuits with beetroot powder baked at different temperatures (**A1**, **B1**, **C1**, **D1** at 150 °C; **A2**, **B2**, **C2**, **D2** at 170 °C) during three time periods.

Polyhenolic Composition mg/100 g	A1	A2	B1	B2	C1	C2	D1	D2
Start of storage								
Gallic acid	2.76 ± 1.11 ^A,1^	2.97 ± 0.12 ^A;1^	22.19 ± 1.06 ^A;2^	15.15 ± 0.69 ^A;2^	25.16 ± 1.20 ^A;3^	31.02 ± 1.45 ^A;3^	32.04 ± 1.56 ^A;4^	41.51 ± 2.16 ^A;4^
Protocatechuic acid	3.33 ± 0.09 ^A;1^	5.11 ± 0.28 ^A;1^	21.11 ± 1.01 ^A;2^	13.51 ± 0.64 ^A;2^	18.16 ± 0.83 ^A;3^	29.73 ± 1.45 ^A;3^	24.86 ± 1.12 ^A;4^	40.36 ± 2.06 ^A;4^
Epicatechin	n.d.	n.d.	n.d.	0.71 ± 0.02 ^1^	2.20 ± 0.10 ^1^	2.07 ± 0.08 ^2^	3.56 ± 0.14 ^2^	n.d.
Catechin	n.d.	n.d.	n.d.	n.d.	n.d.	n.d.	n.d.	n.d.
*p*-Coumaric acid	n.d.	n.d.	n.d.	n.d.	n.d.	n.d.	n.d.	n.d.
Caffeic acid	0.14 ± 0.01 ^A;1^	0.19 ± 0.01 ^A;1^	1.37 ± 0.05 ^A;2^	0.85 ± 0.03 ^A;2^	1.27 ± 0.09 ^A;2^	2.32 ± 0.06 ^A;3^	6.29 ± 0.29 ^A;3^	2.71 ± 0.09 ^A;4^
Chlorogenic acid	n.d	0.14 ± 0.01 ^A;1^	5.29 ± 0.21 ^A;1^	1.45 ± 0.06 ^A;2^	4.28 ± 0.19 ^A;2^	4.22 ± 0.15 ^A;3^	2.25 ± 0.09 ^A;3^	4.33 ± 0.20 ^A;3^
Vanillic acid	13.54 ± 0.61 ^A;1^	11.74 ± 0.43 ^A;1^	24.45 ± 1.23 ^A;2^	11.12 ± 0.51 ^A;1^	25.85 ± 1.11 ^A;2^	18.03 ± 0.90 ^A;2^	20.83 ± 0.98 ^A;3^	21.22 ± 0.82 ^A;3^
Total	19.77	20.15	74.40	42.80	76.92	87.40	89.82	110.13
**3 months**								
Gallic acid	4.26 ± 0.17 ^B;1^	5.52 ± 0.23 ^B;1^	18.34 ± 0.89 ^B;2^	20.00 ± 0.99 ^B;2^	23.73 ± 1.16 ^B;3^	23.69 ± 1.01 ^B;3^	23.36 ± 1.13 ^B;3^	24.02 ± 1.95 ^B;3^
Protocatechuic acid	7.86 ± 0.42 ^B;1^	7.15 ± 0.27 ^B;1^	10.88 ± 0.51 ^B;2^	14.79 ± 0.07 ^B;2^	16.68 ± 0.76 ^B;3^	13.81 ± 0.56 ^B;2^	19.97 ± 1.09 ^B;4^	19.38 ± 0.87 ^B;3^
Epicatechin	n.d.	n.d.	n.d.	n.d.	n.d.	n.d.	n.d.	n.d.
Catechin	n.d.	n.d.	1.54 ± 0.06 ^A;1^	0.18 ± 0.01 ^A;1^	7.46 ± 0.28 ^A;2^	4.88 ± 0.21^A;2^	13.09 ± 0.54 ^A;3^	4.48 ± 0.21 ^A;2^
*p*-Coumaric acid	n.d.	n.d.	0.60 ± 0.03 ^1^	n.d.	0.77 ± 0.02 ^2^	n.d.	0.47 ± 0.01 ^3^	n.d.
Caffeic acid	0.97 ± 0.03 ^B;1^	0.14 ± 0.01 ^B;1^	1.97 ± 0.08 ^B;2^	1.71 ± 1.07 ^B;2^	0.29 ± 0.01 ^B;3^	2.45 ± 0.09 ^B;3^	2.36 ± 0.11	2.32 ± 0.07 ^B;3^
Chlorogenic acid	0.22 ± 0.01 ^A;1^	1.11 ± 0.04 ^B;1^	1.74 ± 0.05 ^B;2^	4.75 ± 0.19 ^B;2^	1.41 ± 0.04 ^B;3^	3.26 ± 0.14 ^B;3^	1.82 ± 0.08 ^B;4^	4.51 ± 0.20 ^B;2^
Vanillic acid	6.24 ± 0.29 ^B;1^	6.24 ± 0.61 ^B;1^	18.02 ± 0.86 ^B;2^	18.00 ± 0.84 ^B;2^	19.19 ± 0.93 ^B;2^	20.60 ± 0.91 ^B;3^	25.86 ± 1.17 ^B;3^	24.68 ± 1.15 ^B;4^
Total	19.56	20.17	53.09	59.44	69.53	68.69	86.93	79.39
**6 months**								
Gallic acid	2.41 ± 0.11 ^C;1^	2.67 ± 0.21 ^C;1^	21.16 ± 0.79 ^C;2^	23.45 ± 0.86 ^C;2^	23.79 ± 1.15 ^C;3^	28.39 ± 1.03 ^C;3^	26.59 ± 1.09 ^C;4^	32.26 ± 1.84 ^C;4^
Protocatechuic acid	3.26 ± 0.13 ^C;1^	3.76 ± 0.28 ^C;1^	17.32 ± 0.48 ^C;2^	16.09 ± 0.06 ^C;2^	20.00 ± 0.66 ^C;3^	25.79 ± 0.58 ^C;3^	20.35 ± 1.07 ^C;3^	28.39 ± 0.44 ^C;3^
Epicatechin	n.d.	n.d.	n.d.	n.d.	n.d.	n.d.	n.d.	n.d.
Catechin	0.86 ± 0.02 ^1^	1.36 ± 0.08 ^1^	2.19 ± 0.04 ^B;2^	1.21 ± 0.02 ^B;2^	2.26 ± 0.06 ^B;2^	1.93 ± 0.11 ^B;3^	7.67 ± 0.10 ^B;3^	6.88 ± 0.43 ^B;4^
*p*-Coumaric acid	n.d.	n.d.	n.d.	n.d.	n.d.	n.d.	n.d.	n.d.
Caffeic acid	1.82 ± 0.06 ^C;1^	0.30 ± 0.02 ^C;1^	2.20 ± 0.06 ^C;2;3^	2.47 ± 1.05 ^C;2^	2.05 ± 0.04 ^C;2^	2.79 ± 0.08 ^C;3^	2.36 ± 0.11 ^C;3^	2.60 ± 0.06 ^C;2;3^
Chlorogenic acid	0.11 ± 0.01 ^B;1^	0.11 ± 0.01 ^A;1^	5.87 ± 0.15 ^C;2^	2.98 ± 0.17 ^C;2^	6.60 ± 0.11 ^C;3^	3.08 ± 0.13 ^C;2^	5.23 ± 0.10 ^C;4^	3.96 ± 0.20 ^C;3^
Vanillic acid	11.23 ± 0.54 ^C;1^	9.98 ± 0.88 ^C;1^	10.84 ± 0.45 ^C;1^	8.27 ± 0.64 ^C;2^	12.97 ± 0.44 ^C;2^	8.19 ± 0.41 ^C;2^	14.98 ± 0.98 ^C;3^	3.91 ± 0.24 ^C;3^
Total	19.69	18.18	59.59	54.51	67.75	70.18	78.14	78.13

Significant differences in mean values (*p* < 0.05) within one type of biscuits between three time periods are presented with different superscript-letters (A, B, C). Significant differences in mean values between four types of biscuits are presented with different superscript numbers (1, 2, 3, 4). n.d.—not detected.

**Table 7 foods-12-00322-t007:** Changes in visual color parameters in biscuit samples baked at different temperatures (**A1**, **B1**, **C1**, **D1** at 150 °C; **A2**, **B2**, **C2**, **D2** at 170 °C) during three time periods.

		Start of Storage			3 Months			6 Months	
	L*	a*	b*	L*	a*	b*	L*	a*	b*
**A1**	53.91 ± 0.05 **^A;3^**	9.05 ± 0.35 **^A;1^**	25.48 ± 0.96 **^A;5^**	55.91 ± 1.58 **^B;4^**	9.45 ± 0.38 **^A;1^**	24.73 ± 0.54 **^A;3^**	56.74 ± 1.34 **^C;4^**	9.22 ± 0.42 **^A;1^**	24.32 ± 0.40 **^A;3^**
**B1**	26.80 ± 0.36 **^A;1^**	23.70 ± 0.88 **^A;4^**	10.62 ± 0.40 **^A;2^**	27.72 ± 1.13 **^A;3^**	24.00 ± 1.55 **^A;2^**	10.49 ± 1.01 **^A;2^**	30.55 ± 1.32 **^B;3^**	26.60 ± 1.29 **^B;2^**	11.59 ± 1.11 **^B;2^**
**C1**	26.57 ± 1.39 **^A;1^**	24.49 ± +0.87 **^A;4^**	11.51 ± 0.62 **^C;3^**	26.23 ± 0.90 **^A;2,3^**	26.12 ± 0.74 **^B;3^**	8.54 ± 0.46 **^A;1^**	27.46 ± 2.12 **^A;1,2^**	26.08 ± 3.29 **^B;2^**	9.80 ± 0.95 **^B;1^**
**D1**	25.88 ± 0.48 **^A;1^**	22.13 ± 2.17 **^A;3^**	9.94 ± 0.76 **^B;1^**	25.47 ± 1.30 **^A;2^**	24.36 ± 0.69 **^B;2^**	9.14 ± 0.56 **^A;1^**	27.16 ± 0.89 **^B;1,2^**	24.49 ± 0.84 **^B;2^**	9.97 ± 0.63 **^B;1^**
**A2**	53.14 ± 0.33 **^A;3^**	10.04 ± 0.49 **^B;2^**	27.14 ± 1.17 **^B;6^**	56.82 ± 1.03 **^C;4^**	9.13 ± 0.53 **^A;1^**	24.41 ± 0.47 **^A;3^**	55.61 ± 0.83 **^B;4^**	9.49 ± 0.37 **^A;1^**	26.09 ± 2.04 **^B;3^**
**B2**	28.83 ± 0.76 **^B;2^**	23.99 ± 0.52 **^A;4^**	13.42 ± 0.58 **^B;4^**	26.67 ± 0.78 **^A;2,3^**	26.33 ± 0.64 **^B;3^**	10.68 ± 0.77 **^A;2^**	28.44 ± 1.27 **^B;2^**	26.81 ± 1.54 **^B;2^**	10.88 ± 1.75 **^A;2^**
**C2**	25.99 ± 0.98 **^A;1^**	22.22 ± 1.21 **^A;3^**	10.73 ± 1.13 **^A;2^**	25.22 ± 1.82 **^A;2^**	24.80 ± 1.66 **^B;2^**	10.21 ± 1.59 **^A;2^**	26.58 ± 1.26 **^A;1,2^**	25.01 ± 2.45 **^C;2^**	10.14 ± 1.54 **^A;2^**
**D2**	25.21 ± 0.51 **^B;1^**	21.32 ± 1.62 **^A;3^**	9.05 ± 0.98 **^A;1^**	24.55 ± 0.83 **^A;1^**	24.76 ± 0.95 **^B;2^**	9.49 ± 1.25 **^AB;1,2^**	25.88 ± 0.58 **^B;1^**	25.75 ± 0.87 **^B;2^**	10.69 ± 1.22 **^B;2^**

Significant differences in mean values (*p* < 0.05) within one type of biscuits between three time periods are presented with different superscript-letters (A, B, C). Significant differences in mean values between four types of biscuits are presented with different superscript-numbers (1, 2, 3, etc.).

**Table 8 foods-12-00322-t008:** Water activity (*aw*).

	Start of Storage	3 Months
	*aw*	*aw*
**A1**	0.429 ± 0.006	0.355 ± 0.000
**A2**	0.327 ± 0.000	0.356 ± 0.001
**B1**	0.559 ± 0.001	0.349 ± 0.001
**B2**	0.387 ± 0.002	0.350 ± 0.000
**C1**	0.501 ± 0.004	0.347 ± 0.003
**C2**	0.415 ± 0.005	0.344 ± 0.001
**D1**	0.450 ± 0.002	0.335 ± 0.004
**D2**	0.379 ± 0.001	0.341 ± 0.001

## Data Availability

Data are contained within the article.

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
