# Peer review of "Effect of Beetroot Powder Incorporation on Functional Properties and Shelf Life of Biscuits"

_foods, 2023, doi:10.3390/foods12020322_

Round 1

Reviewer 1 Report

kindly provide the HPLC chromatogram  showed the presence of epicatechin, catechin, gallic, protocatechinic, p-coumarinic, caffeic, chlorogenic, and vanillic acid.

kindly add in discussion about  reason in change in biscuits' color a during storage. 

Author Response

Dear Reviewer,

Thank you for your constructive suggestions in order to improve the quality of our manuscript. Please find the response to your comments in the pdf document attached via the MDPI system.

Best wishes,

Prof. Dr. Nebojsa Pantelic

Reviewer 2 Report

GENERAL COMMENT TO EDITOR

This study aimed to evaluate the nutritional and functional profile of biscuits incorporating beetroot power. The research presented is well done but some parts of the discussion need to be revised. Also, the statistical analysis should be presented in all Figures and Tables.

Please, see my comments below:

Lines 39-42 and 53-55: Sentences should be supported by a reference

Line 45: please, give information about the low temperatures usually applied

Line 67: give some examples about the different forms of beetroot ingredient used in the bakery sector

Line 121: please, give information about the ºBrix and moisture content of beetroot samples and obtained powders

Line 123: Did authors want to mention “dried”?? instead of “baked”?

Line 189: please, indicate the sampling of each batch formulation.

Line 342: Authors stated that “High water binding capacities (2.00 g for BP 342 and 7.12 g for SF), as well as oil binding capacities (3.56 g for BP and 3.42 g for SF) were found.” These results are compared to which samples? Please, clarify. Also, results suggest no statistical differences for OBC between BP and SF!

Table 2: should include the abbreviation of BP and SF to help the discussion. Also, it may include the statistical analysis.

Lines 337 e 379: information about the ash results is repeated. Please, reframe

Table 3: title should include “biscuit samples prepared at 150 ºC”. Once again, the statistical analysis should be included.

Lines 339-404: this section belongs to introduction

Line 418: “The temperature is factor that affecting the stability of betalains during storage”- needs English edition

 Lines 465-467: Please, include values of polyphenols content to help in the discussion 

Line 474: “The reason for this could be the inhomogeneity of the ingredients in the biscuits”- please, explore the idea. How many biscuits were made for each formulation? The SD are relatively low.

Lines 542-546: In my opinion, the food matrices are very different to extrapolate this type of observation.

Line 605: “Polyphenols were lost by 14% after six months of storage”. Are authors referring to SF? Please, clarify

Lines 626-629: Any explanation for this observation or mentioned by [61] for barley varieties (and corresponding malts)? 

Lines 647-651. Why is this information included? Authors should explore and connect the reported literatures with the results.

Author Response

(The authors gave the same response as above.)

Reviewer 3 Report

(1)      Your title “Effect of beetroot powder incorporation on functional properties and shelf life of biscuits” What are the functional properties?

(2)      The abstract needs to provide a specific value of the desired result

(3)      Many formatting issues, such as line-158, too many spaces. Many paragraphs have spaces at the beginning, but many do not have spaces and need to be consistent.

(4)      Does beetroot powder have its own color? Does it affect the color of the biscuits?

(5)      What do you think is the reason why there are still antioxidants after high temperature roasting and why the structure of these active substances is not destroyed?

(6)      Why not to determine the flavor, and texture, the substance added will affect the taste?

(7)      The figure should be labeled as significant.

(8)      The conclusion is so poorly written it needs focus

(9)      Writing expression needs further improvement

Author Response

(The authors gave the same response as above.)

Reviewer 4 Report

The topic of the research article is of great interest. However, it requires some improvement. The main drawbacks of this manuscript Below are several specific comments:

1. The whole manuscript is mixed between American English and British English, the authors even use British English or American English. For consistency, consider replacing it with the American English spelling.

2. Check the names of the compounds (protocatechinic, p-coumarinic, etc.), in the whole manuscript

3. Spacing of numbers and units

4. For the statical analysis, the authors should apply the one-way ANOVA using the Post Hoc Multiple comparisons using the equal variances assumed by Tukey or Duncan. That can help the reader to identify the difference between the batches by the superscripts litters to know the significantly different between them.

5. Minor points

Line 25 adds a comma before and                                                                                                         

Line 39 adds a comma before and

Line 41 adds a or the before dark

line 55 changes speed up metabolism, and enhance to speeding up metabolism and enhancing

line 76 adds more before vitamins

line 85 adds of before the

line 88 adds the before addition

line 93 adds a comma before and

line 100 adds a comma before and                                                                                                                   

line 100 changes flavonoids to flavonoid

line 101 changes was to were

line 102 changes was to were

line 108 is it protocatechinic, p-coumarinic caffeic, or protocatechuic, p-coumaric, caffeic

line 110 is it tetramethylchroman or tetramethylchromane

Line 121 changes were to was

line 146 adds a before gram

Line 159 changes were to was

Line 159 changes that adhering particles to those adhering particles or that adhering particle

Line 174 adds the before visual

Line 203 adds a comma before and

line 207 changes were to was

line 207 adds the before hydrolysis

line 210 changes which to that

line 219 changes heat stable to heat-stable

line 220 adds a comma before and

line 223 adds a comma before and

line 225 adds a comma before and

Line 229 changes Afterwards to Afterward

line 249 changes an to a

Line 273 adds the before antioxidant

Line 277 changes Afterwards to Afterward

Line 281 changes ferric reducing to ferric-reducing

Line 282 adds the before FRAP

Line 295 changes precolumn to pre-column

Line 310 changes bacteriacea to bacteriaceae

Line 316 adds the before sample

Line 317 adds a comma before and

Line 317 adds and before standard              

Line 319 changes a non-parametric analogs to a non-parametric analog

Line 320 adds the before significance

line 321 removes period of

line 323 adds the before test and changes was to were

Line 328 adds of before the

line 337 adds a comma before and

line 346 adds a comma before and

line 360 changes For to  To

Line 361 adds were before prepared

Line 389, 391, and 392 add a comma before and

Line 413 changes were to was

Line 415 adds a comma after content

Line 416 changes amount to number

Line 418 adds a or the before factor

Line 444 changes rich sources to a rich source

Line 448 changes polyphenol to polyphenols

line 477 adds a comma before and

line 482 adds were before prepared

line 484 adds a before TPC

line 486 change the portion to a portion

line 503 changes sample to samples

line 506 add a comma before and

line 525 removes are and change showing to show

Line 529 removes an

Line 533 removes the comma after sample

Line 534 changes Figure to Figures

line 532 changes the to a

line 533 adds a comma before and

Line 562 and 564 change ferro to Ferro

Line 578 changes temperatures to temperature

Line 578 adds the before start

Lines 593 to 600 check the names of the compounds

Lines 611 and 616 adds a comma before and

line 604 remove the comma after powder

line 628 changes like in to like to

line 633 removes the comma after storage

Line 654 adds a or the before consumer

Line 656 changes was to were

Line 658 adds a comma before and

line 667 changes beeroot to beetroot

Line 733 adds a comma before and aerobic

Line 742 removes are

Line 744 removes that

Line 746 changes was to were

Line 749 changes sample to samples

line 751 adds by after Research

line 755 changes sample to samples

line 781 changes its to their

line 783 change darker colored to darker-colored

line 784 adds a comma before and

line 784 adds a comma before and

Author Response

(The authors gave the same response as above.)

Round 2

Reviewer 3 Report

After my evaluation, it could be accepted in this form. All questions all well answered.